# Synthesis and Physicochemical Characterization of Novel Dicyclopropyl-Thiazole Compounds as Nontoxic and Promising Antifungals

**DOI:** 10.3390/ma14133500

**Published:** 2021-06-23

**Authors:** Anna Biernasiuk, Anna Banasiewicz, Maciej Masłyk, Aleksandra Martyna, Monika Janeczko, Angelika Baranowska-Łączkowska, Anna Malm, Krzysztof Z. Łączkowski

**Affiliations:** 1Department of Pharmaceutical Microbiology, Faculty of Pharmacy, Medical University of Lublin, Chodźki 1, 20-093 Lublin, Poland; anna.malm@umlub.pl; 2Department of Chemical Technology and Pharmaceuticals, Faculty of Pharmacy, Collegium Medicum, Nicolaus Copernicus University, Jurasza 2, 85-089 Bydgoszcz, Poland; banasiewicz.anna@o2.pl (A.B.); krzysztof.laczkowski@cm.umk.pl (K.Z.Ł.); 3Department of Molecular Biology, Faculty of Science and Health, The John Paul II Catholic University of Lublin, Konstantynów 1i, 20-708 Lublin, Poland; maciekm@kul.pl (M.M.); aleksandra.martyna@kul.pl (A.M.); mjanec@kul.pl (M.J.); 4Institute of Physics, Kazimierz Wielki University, Powstańców Wielkopolskich 2, 85-090 Bydgoszcz, Poland; angelika.baranowska@ukw.edu.pl

**Keywords:** antifungal activity, *Candida albicans*, antibiofilm effect, mode of action, cytotoxicity, haemolytic assay, HOMO–LUMO, molecular electrostatic potential

## Abstract

There is a need to search for new antifungals, especially for the treatment of the invasive *Candida* infections, caused mainly by *C. albicans*. These infections are steadily increasing at an alarming rate, mostly among immunocompromised patients. The newly synthesized compounds (**3a**–**3k**) were characterized by physicochemical parameters and investigated for antimicrobial activity using the microdilution broth method to estimate minimal inhibitory concentration (MIC). Additionally, their antibiofilm activity and mode of action together with the effect on the membrane permeability in *C. albicans* were investigated. Biofilm biomass and its metabolic activity were quantitatively measured using crystal violet (CV) staining and tetrazolium salt (XTT) reduction assay. The cytotoxic effect on normal human lung fibroblasts and haemolytic effect were also evaluated. The results showed differential activity of the compounds against yeasts (MIC = 0.24–500 µg/mL) and bacteria (MIC = 125–1000 µg/mL). Most compounds possessed strong antifungal activity (MIC = 0.24–7.81 µg/mL). The compounds **3b**, **3c** and **3e**, showed no inhibitory (at 1/2 × MIC) and eradication (at 8 × MIC) effect on *C. albicans* biofilm. Only slight decrease in the biofilm metabolic activity was observed for compound **3b**. Moreover, the studied compounds increased the permeability of the membrane/cell wall of *C. albicans* and their mode of action may be related to action within the fungal cell wall structure and/or within the cell membrane. It is worth noting that the compounds had no cytotoxicity effect on pulmonary fibroblasts and erythrocytes at concentrations showing anticandidal activity. The present studies in vitro confirm that these derivatives appear to be a very promising group of antifungals for further preclinical studies.

## 1. Introduction

In the last years, the prevalence of *Candida* infections, which range from superficial to deep-seated invasive candidiasis, has increased at an alarming rate, especially among immunocompromised individuals [1,2]. Given the fact that at least 50% of healthy individuals are the carriers of *Candida* species, these commensal microorganisms are regarded as potentially pathogenic in susceptible hosts. Infections caused by *Candida* spp. are primarily dependent on the immunological status of the host. Both local and systemic risk factors may result in weakened immune functions that mediate *Candida* colonization on host surfaces. These fungi mostly affect neutropenic patients, individuals with haematological malignancies, older patients, diabetics or individuals pre-exposed to azoles or echinocandins [1,2,3,4,5]. The most common isolated *Candida* species of clinical significance in fungal invasive infection is *Candida albicans*. It is associated with the occurrence of mortality rates as high as 35–50% [6,7]. Non-*albicans Candida* spp. (NAC), such as *C. glabrata, C. krusei, C. tropicalis* or *C. parapsilosis*, are also serious nosocomial threats. An important factor of *Candida* pathogenicity is the ability to form a biofilm. The biofilm structure can be observed both on the surfaces of different host tissues and various medical devices (such as central venous catheters, prosthetic heart valves, contact lenses, urinary catheters or dentures). It protects yeast cells from the immune system and increases the resistance of *Candida* strains to conventional antifungals, due to the difficulty of penetration into the extracellular matrix. Yeasts in the biofilm community are more resistant and can survive up to 1000 times higher concentrations of antimycotics than free-living cells [3,8,9,10,11]. Currently, the list of the commercially available antifungal agents, used for the treatment of infections caused by *Candida* spp. is limited to three major classes: polyenes (e.g., amphotericin B or nystatin), azoles (e.g., fluconazole or posaconazole) and echinocandins (e.g., caspofungin or micafungin) [6,7]. The treatment of fungal infections is often ineffective and there is a need to search for new antifungals.

The chemistry of thiazoles is currently widely developed due to the relatively easy synthesis of a wide range of compounds, which enables studying the relationship between their structure and properties [12,13]. Due to the nonlinear optical properties (NLO) of the thiazole system, it is used in optical devices such as photovoltaic detectors or light-emitting diodes [14]. However, its greatest advantage is that it exhibits a wide range of biological properties, such as antimicrobial [15,16,17,18], anticancer [19,20,21], anticonvulsant [22], antioxidant [23] and anti-SARS-CoV [24]. Numerous studies, including those carried out over the years by our group, show that thiazole derivatives have high antifungal activity [3,25]. The cyclopropane system, due to its unique properties, such as low molecular weight, conformational rigidity, unusual bonding and planarity, is used in drug design as a replacement for gem-dimethyl and alkene groups and phenyl rings, leading to the reduced lipophilicity and increased metabolic stability of drugs [26,27]. We have recently shown that thiazole derivatives containing a cyclopropane ring exhibit high activity against both reference strains and clinical isolates of *Candida* spp. with minimal inhibitory concentration (MIC) ranging from 0.008 to 7.81 µg/mL comparable or even higher than that of nystatin [28,29]. Additionally, the designed derivatives showed lower toxicity compared to nystatin.

The promising results prompted us to synthesize thiazole derivatives containing two cyclopropyl groups and then to investigate their antifungal activity against *Candida* spp.

A better understanding of the intermolecular interactions involving drug molecules and their reactivity is possible by applying modern quantum chemistry methods. Here, a widely employed example includes approximations based on the density functional theory (DFT). In the present study, the DFT is used to find the optimal geometrical parameters of the investigated compounds as well as to evaluate the shape and the energy of their highest occupied molecular orbital (HOMO) and the lowest unoccupied molecular orbital (LUMO). These are the two main orbitals involved in chemical reactions, with the HOMO playing the role of an electron donor and the LUMO of an electron acceptor. Within the present work, their energies are denoted as *E_HOMO_* and *E_LUMO_*, respectively. The HOMO–LUMO energy gap Δ*E* is defined as
Δ*E* = *E_LUMO_* − *E_HOMO_*,
and gives insight into the molecule’s activity. Low values of Δ*E* result in higher chemical activity and lower kinetic stability of the compound, while high Δ*E* values result in lower chemical activity and higher kinetic stability. The knowledge on the HOMO and LUMO energy values allows us to evaluate the electron affinity (*EA)* and ionization potential *IP* and the molecule’s hardness *η*, softness *S*, and electronegativity *χ*.

The molecule’s *EA* provides information about the molecule’s reductant properties and is related to the LUMO energy:*EA* = −*E_LUMO_*.

By definition, *EA* is the energy released upon a molecule gaining an electron. The *IP* is the energy needed to remove an electron from a molecule. Thus, a lower *IP* means an easier electron removal, that is, a greater tendency for the molecule to donate electrons. Its value is related to the HOMO energy as
*IP* = −*E_HOMO_*.

The molecule’s electronegativity *χ* can instead be defined as [30]
*χ* = (*IP* + *EA*)/2,
and represents its ability to attract electrons. The molecule’s hardness *η* is the inverse of the molecule’s softness *S* and is defined as [30]
*η* = 1/*S* = (*IP* − *EA*)/2.

The larger the value of *η*, the larger the molecule’s resistance to changing the number of electrons. Additionally, within the present work, the molecular electrostatic potential (MEP) surfaces are calculated for the investigated systems. They allow molecular sites susceptible to nucleophilic or electrophilic attacks to be indicated.

In the present work, novel dicyclopropyl-thiazole compounds were designed, synthesized and characterized by physicochemical parameters, and they were investigated in terms of antimicrobial activity. Additionally, we conducted a screening study of their antibiofilm activity together with their mode of action and their effect on the membrane permeability in *C. albicans*. We also evaluated the cytotoxic effect on normal human lung fibroblasts and erythrocytes.

## 2. Materials and Methods

### 2.1. Chemistry

All substrates were bought from Sigma-Aldrich Chemicals, St. Louis, MO, USA, and experiments were performed under an air atmosphere. The ^1^H (700 MHz) and ^13^C (176 MHz) NMR analyses were carried out employing a Bruker Avance III multinuclear spectrometer. The high-resolution mass spectra were recorded by the Laboratory for Analysis of Organic Compounds and Polymers, Polish Academy of Science, Łódź. Open glass capillaries were used to determine melting point values, and the results are uncorrected. The Macherey-Nagel Polygram Sil G/UV254 0.2 mm plates were employed in thin-layer chromatography (TLC) analyses.

#### 2.1.1. Synthesis and Structural Characterization

##### 2-(Dicyclopropylmethylene)hydrazinecarbothioamide (**1**)

To a stirred solution of dicyclopropyl ketone (**1**) (2.20 g, 20.0 mmol) in absolute ethyl alcohol (30 mL), thiosemicarbazide (1.82 g, 20.0 mmol) was added, followed by the addition (1.0 mL) of acetic acid. After stirring under reflux for 20 h, the reaction mixture was poured into water (50 mL) and neutralized with sodium bicarbonate solution. After extraction with dichloromethane (2 × 100 mL), the solvent was evaporated in vacuo to produce a product with yield 2.14 g (56%); mp 80–85 °C; eluent: dichloromethane/methanol (95:5), R_f_ = 0.67. ^1^H NMR (DMSO-d_6_, 700MHz); δ (ppm): 0.54–0.61 (m, 2H, CH_2_); 0.73–0.78 (m, 2H, CH_2_); 0.84–0.92 (m, 4H, 2CH_2_); 1.13–1.18 (m, 1H, CH); 2.10–2.16 (m, 1H, CH); 7.36 (bs, 1H, NH_2_); 8.00 (bs, 1H, NH_2_); 10.00 (bs, 1H, NH). ^13^C NMR (DMSO-d_6_, 175 MHz); δ(ppm): 6.33 (CH_2_); 7.42 (CH_2_); 10.38 (CH_2_); 11.52 (CH_2_); 11.75 (CH); 20.52 (CH); 157.83 (C=N); 178.62 (C=S).

##### 2-(2-(Dicyclopropylmethylene)hydrazinyl)-4-(4-fluorophenyl)thiazole (**3a**)

Typical procedure Carbothioamide **2** (0.183 g, 1.0 mmol) was added to a stirred solution of 2-bromo-1-(4-fluorophenyl)ethanone (0.217 g, 1.0 mmol) in absolute ethyl alcohol (25 mL). The reaction mixture was stirred under room temperature for 20 h, and the separated precipitate was collected by filtration to produce the desired product with yield 0.16 g, 53%, (dichloromethane, R_f_ = 0.49); mp 135–138 °C. ^1^H NMR (DMSO-d_6_, 700 MHz); δ (ppm): 0.62–0.70 (m, 4H, 2CH_2_); 0.89–0.98 (m, 4H, 2CH_2_); 1.17–1.26 (m, 1H, CH); 2.06–2.15 (m, 1H, CH); 7.19 (s, 1H, CH); 7.20–7.26 (m, 2H, 2CH); 7.83–7.89 (m, 2H, 2CH); 10.98 (bs, 1H, NH). ^13^C NMR (DMSO-d_6_, 175 MHz); δ(ppm): 6.47 (2CH_2_); 7.13 (2CH_2_); 11.24 (CH); 12.18 (CH); 103.78 (CH_thiazole_); 115.94 (2CH_Ar_); 128.18 (2CH_Ar_); 130.67 (C); 147.44 (C); 158.34 (C=N); 162.24 (d, C–F, J = 243 Hz); 170.69 (C-NH). ESI-HRMS (*m*/*z*) calculated for C_16_H_17_FN_3_S: 302.1127 [M + H]^+^. Found: 302.11127 [M + H]^+^.

##### 4-(4-Chlorophenyl)-2-(2-(dicyclopropylmethylene)hydrazinyl)thiazole (**3b**)

Yield: 0.10 g, 32%, (dichloromethane/methanol (95:5), R_f_ = 0.92); mp 146–147 °C. ^1^H NMR (DMSO-d_6_, 700 MHz); δ (ppm): 0.63–0.71 (m, 4H, 2CH_2_); 0.91–0.99 (m, 4H, 2CH_2_); 1.25–1.19 (m, 1H, CH); 2.06–2.12 (m,1H, CH); 7.30 (s, 1H, CH); 7.47 (d, 2H, 2CH, J = 8.6 Hz); 7.81 (d, 2H, 2CH, J = 8.5 Hz); 11.17 (bs, 1H, NH). ^13^C NMR (DMSO-d_6_, 175 MHz); δ(ppm): 6.26 (2CH_2_); 7.09 (2CH_2_); 11.19 (CH); 11.95 (CH); 104.56 (CH_thiazole_); 127.70 (2CH_Ar_); 129.00 (2CH_Ar_); 132.32 (C); 134.34 (C); 149.06 (C); 156.44 (C=N); 170.93 (C–NH). ESI-HRMS (*m*/*z*) calculated for C_16_H_17_ClN_3_S: 318.0832 [M + H]^+^. Found: 318.0831 [M + H]^+^.

##### 4-(4-Bromophenyl)-2-(2-(dicyclopropylmethylene)hydrazinyl)thiazole (**3c**)

Yield: 0.16 g, 44%, (dichloromethane/methanol (95:5), R_f_ = 0.45); mp 139–140 °C. ^1^H NMR (DMSO-d_6_, 700 MHz); δ (ppm): 0.62–0.69 (m, 4H, 2CH_2_); 0.89–0.98 (m, 4H, 2CH_2_); 0.18–1.23 (m, 1H, CH); 2.06–2.12 (m, 1H, CH); 7.30 (s, 1H, CH); 7.59 (d, 2H, 2CH, J = 8.4 Hz); 7.76 (d, 2H, 2CH, J = 8.4 Hz); 11.13 (bs, 1H, NH). ^13^C NMR (DMSO-d_6_, 175 MHz); δ(ppm): 6.34 (2CH_2_); 7.16 (2CH_2_); 11.21 (CH); 12.04 (CH); 104.73 (CH_thiazole_); 121.08 (C); 128.07 (2CH_Ar_); 131.97 (2CH_Ar_); 148.57 (C); 157.18 (C=N); 170.87 (C–NH). ESI-HRMS (*m*/*z*) calculated for C_16_H_17_BrN_3_S: 362.0327 [M + H]^+^. Found: 362.0328 [M + H]^+^.

##### 4-(2-(2-(Dicyclopropylmethylene)hydrazinyl)thiazol-4-yl)benzonitrile (**3d**)

Yield: 0.18 g, 58%, (dichloromethane/methanol (95:5), R_f_ = 0.85); mp 138–140 °C. ^1^H NMR (DMSO-d_6_, 700 MHz); δ (ppm): 0.61–0.68 (m, 4H, 2CH_2_); 0.88–0.97 (m, 4H, 2CH_2_); 1.17–1.23 (m, 1H, CH); 2.07–2.13 (m, 1H, CH); 7.52 (s, 1H, CH); 7.84 (d, 2H, 2CH, J = 8.5 Hz); 8.00 (d, 2H, 2CH, J = 6.9 Hz); 11.08 (bs, 1H, NH). ^13^C NMR (DMSO-d_6_, 175 MHz); δ(ppm): 6.30 (2CH_2_); 7.14 (2CH_2_); 11.18 (CH); 12.01 (CH); 107.64 (CH_thiazole_); 109.99 (C); 119.41 (C); 126.60 (2CH_Ar_); 133.07 (2CH_Ar_); 139.27 (C); 148.69 (C); 156.83 (C=N); 171.11 (C–NH). ESI-HRMS (*m*/*z*) calculated for C_17_H_17_N_4_S: 309.1174 [M + H]^+^. Found: 309.1181 [M + H]^+^.

##### 2-(2-(Dicyclopropylmethylene)hydrazinyl)-4-*p*-tolylthiazole (**3e**)

Yield: 0.16 g, 53%, (dichloromethane, R_f_ = 0.32); mp 128–131 °C. ^1^H NMR (DMSO-d_6_, 700 MHz); δ (ppm): 0.63–0.72 (m, 4H, 2CH_2_); 0.90–1.00 (m, 4H, 2CH_2_); 0.18–1.27 (m, 1H, CH); 2.06–2.15 (m, 1H, CH); 2.32 (s, 3H, CH_3_); 7.14 (s, 1H, CH); 7.21 (d, 2H, 2CH, J = 7.9 Hz); 7.71 (d, 2H, 2CH, J = 8.1 Hz); 10.96 (bs, 1H, NH). ^13^C NMR (DMSO-d_6_, 175 MHz); δ(ppm): 6.57 (2CH_2_); 7.45 (2CH_2_); 11.32 (CH); 12.34 (CH); 21.27 (CH_3_); 103.20 (CH_thiazole_); 126.12 (2CH_Ar_); 129.72 (2CH_Ar_); 130.61 (C); 138.02 (C); 147.28 (C); 159.50 (C=N); 170.46 (C–NH). ESI-HRMS (*m*/*z*) calculated for C_17_H_20_N_3_S: 298.1378 [M + H]^+^. Found: 298.1380 [M + H]^+^.

##### 4-(4-Azidophenyl)-2-(2-(dicyclopropylmethylene)hydrazinyl)thiazole (**3f**)

Yield: 0.12 g, 65%, (dichloromethane, R_f_ = 0.34); mp 144–147 °C. ^1^H NMR (DMSO-d_6_, 700 MHz); δ (ppm): 0.61–0.68 (m, 4H, 2CH_2_); 0.88–0.97 (m, 4H, 2CH_2_); 1.17–1.23 (m, 1H, CH); 2.06–2.13 (m, 1H, CH); 7.14 (d, 2H, 2CH, J = 6.5 Hz); 7.20 (s, 1H, CH); 7.86 (d, 2H, 2CH, J = 8.5 Hz); 11.01 (bs, 1H, NH). ^13^C NMR (DMSO-d_6_, 175 MHz); δ(ppm): 6.37 (2CH_2_); 7.21 (2CH_2_); 11.23 (CH); 12.06 (CH); 103.78 (CH_thiazole_); 119.81 (2CH_Ar_); 127.68 (2CH_Ar_); 129.43 (C); 138.96 (C); 148.50 (C); 157.48 (C=N); 170.75 (C–NH). ESI-HRMS (*m*/*z*) calculated for C_16_H_17_N_6_S: 325.1235 [M + H]^+^. Found: 325.1238 [M + H]^+^.

##### 2-(2-(Dicyclopropylmethylene)hydrazinyl)-4-(4-(trifluoromethyl)phenyl)thiazole (**3g**)

Yield: 0.21 g, 61%, (dichloromethane, R_f_ = 0.58); mp 157–159 °C. ^1^H NMR (DMSO-d_6_, 700 MHz); δ (ppm): 0.61–0.67 (m, 4H, 2CH_2_); 0.88–0.97 (m, 4H, 2CH_2_); 1.17–1.23 (m, 1H, CH); 2.07–2.13 (m, 1H, CH); 7.45 (s, 1H, CH); 7.74 (d, 2H, 2CH, J = 8.3 Hz); 8.04 (d, 2H, 2CH, J = 8.1 Hz); 11.05 (bs, 1H, NH). ^13^C NMR (DMSO-d_6_, 175 MHz); δ(ppm): 6.29 (2CH_2_); 7.13 (2CH_2_); 11.18 (CH); 12.00 (CH); 106.57 (CH_thiazole_); 124.05 (C); 125.98 (2CH_Ar_); 126.53 (2CH_Ar_); 127.99 (q, C, J_C-F_ = 31.7 Hz); 138.81 (C); 148.72 (C); 156.77 (C=N); 171.08 (C–NH). ESI-HRMS (*m*/*z*) calculated for C_17_H_17_F_3_N_3_S: 352.1095 [M + H]^+^. Found: 352.1096 [M + H]^+^.

##### 2-(2-(Dicyclopropylmethylene)hydrazinyl)-4-(4-nitrophenyl)thiazole (**3h**)

Yield: 0.32 g, 99%, (dichloromethane, R_f_ = 0.33); mp 141–143 °C. ^1^H NMR (DMSO-d_6_, 700 MHz); δ (ppm): 0.63–0.70 (m, 4H, 2CH_2_); 0.89–1.00 (m, 4H, 2CH_2_); 1.17–1.26 (m, 1H, CH); 2.07–2.17 (m, 1H, CH); 7.61 (s, 1H, CH); 8.10 (d, 2H, 2CH, J = 9.1 Hz); 8.27 (d, 2H, 2CH, J = 9.1 Hz); 11.08 (bs, 1H, NH). ^13^C NMR (DMSO-d_6_, 175 MHz); δ(ppm00): 6.30 (2CH_2_); 7.13 (2CH_2_); 11.17 (CH); 12.00 (CH); 108.67 (CH_thiazole_); 124.50 (2CH_Ar_); 126.78 (2CH_Ar_); 141.30 (C); 146.62 (C); 148.55 (C); 156.73 (C=N); 171.23 (C–NH). ESI-HRMS (*m*/*z*) calculated for C_16_H_17_N_4_O_2_S: 329.1072 [M + H]^+^. Found: 329.1071 [M + H]^+^.

##### 3-Chloro-*N*-(4-(2-(2-(dicyclopropylmethylene)hydrazinyl)thiazol-4-yl)phenyl)propanamide (**3i**)

Yield: 0.18 g, 46%, (dichloromethane/methanol (95:5), R_f_ = 0.52); mp 210–213 °C. ^1^H NMR (DMSO-d_6_, 700 MHz); δ (ppm): 0.62–0.70 (m, 4H, 2CH_2_); 0.90–0.98 (m, 4H, 2CH_2_); 1.19–1.25 (m, 1H, CH); 2.06–2.12 (m, 1H, CH); 2.83 (t, 2H, 2CH, J = 6.1 Hz); 3.88 (t, 2H, 2CH, J = 6.2 Hz); 7.11 (s, 1H, CH); 7.64 (d, 2H, 2CH, J = 8.5 Hz); 7.75 (d, 2H, 2CH, J = 8.6 Hz); 10.25 (bs, 1H, NH); 11.03 (bs, 1H, NH). ^13^C NMR (DMSO-d_6_, 175 MHz); δ(ppm): 6.40 (2CH_2_); 7.32 (2CH_2_); 11.35 (CH); 12.16 (CH); 41.21 (CH_2_); 102.63 (CH_thiazole_); 119.67 (2CH_Ar_); 126.49 (2CH_Ar_); 128.81 (C); 139.25 (C); 147.59 (C); 158.38 (C=N); 168.47 (C=O); 170.51 (C–NH). ESI-HRMS (*m*/*z*) calculated for C_19_H_22_ClN_4_OS: 389.1203 [M + H]^+^. Found: 389.1202 [M + H]^+^.

##### 2-Chloro-*N*-(4-(2-(2-(dicyclopropylmethylene)hydrazinyl)thiazol-4-yl)phenyl)acetamide (**3j**)

Yield: 0.14 g, 37%, (dichloromethane/methanol (95:5), R_f_ = 0.83); mp >300 °C. ^1^H NMR (DMSO-d_6_, 700 MHz); δ (ppm): 0.61–0.68 (m, 4H, 2CH_2_); 0.88–0.96 (m, 4H, 2CH_2_); 1.18–1.22 (m, 1H, CH); 2.07–2.13 (m, 1H, CH); 4.26 (s, 2H, CH_2_); 7.12 (s, 1H, CH); 7.62 (d, 2H, 2CH, J = 8.8 Hz); 7.78 (d, 2H, 2CH, J = 7.8 Hz); 10.43 (bs, 1H, NH); 11.03 (bs, 1H, NH). ^13^C NMR (DMSO-d_6_, 175 MHz); δ(ppm): 6.41 (2CH_2_); 7.28 (2CH_2_); 11.32 (CH); 12.15 (CH); 44.02 (CH_2_); 102.91 (CH_thiazole_); 119.93 (2CH_Ar_); 126.55 (2CH_Ar_); 129.54 (C); 138.65 (C); 147.89 (C); 158.15 (C=N); 165.15 (C=O); 170.57 (C–NH). ESI-HRMS (*m*/*z*) calculated for C_18_H_20_ClN_4_OS: 375.1046 [M + H]^+^. Found: 375.1047 [M + H]^+^.

##### 4-(3,4-Dichlorophenyl)-2-(2-(dicyclopropylmethylene)hydrazinyl)thiazole (**3k**)

Yield: 0.27 g, 77%, (dichloromethane, R_f_ = 0.68); mp 152–154 °C. ^1^H NMR (DMSO-d_6_, 700 MHz); δ (ppm): 0.63–0.70 (m, 4H, 2CH_2_); 0.89–0.98 (m, 4H, 2CH_2_); 1.17–1.26 (m, 1H, CH); 2.07–2.16 (m, 1H, CH); 7.43 (s, 1H, CH); 7.64 (d, 2H, 2CH, J = 8.3 Hz); 7.81 (dd, 1H, CH, J_1_ = 1.8 Hz, J_2_ = 8.6 Hz); 11.01 (bs, 1H, NH). ^13^C NMR (DMSO-d_6_, 175 MHz); δ(ppm): 6.28 (2CH_2_); 7.12 (2CH_2_); 11.19 (CH); 11.99 (CH); 106.02 (CH_thiazole_); 126.01 (C); 127.68 (C); 130.07 (C); 131.26 (C); 131.88 (C); 135.73 (C); 147.76 (C); 156.75 (C=N); 171.00 (C–NH). ESI-HRMS (*m*/*z*) calculated for C_16_H_16_Cl_2_N_3_S: 352.0442 [M + H]^+^. Found: 352.0444 [M + H]^+^.

### 2.2. Microorganisms

The reference strains of microorganisms from American Type Culture Collection (ATCC), Manassas, VA, USA, were included. The representative Gram-positive bacteria were: *Staphylococcus aureus* ATCC 6538, *Staphylococcus aureus* ATCC 25923, *Staphylococcus epidermidis* ATCC 12228, *Micrococcus luteus* ATCC 10240, *Bacillus subtilis* ATCC 6633 and *Bacillus cereus* ATCC 10876, while those of Gram-negative bacteria: *Escherichia coli* ATCC 25922, *Proteus mirabilis* ATCC 12453, *Klebsiella pneumoniae* ATCC 13883, *Salmonella* Typhimurium ATCC 14028, *Pseudomonas aeruginosa* ATCC 9027 and *Bordetella bronchiseptica* ATCC 4617. Moreover, the fungi belonging to yeasts (*Candida albicans* ATCC 2091, *Candida albicans* ATCC 10231, *Candida parapsilosis* ATCC 22019, *Candida glabrata* ATCC 90030 and *Candida krusei* ATCC 14243) were used.

### 2.3. Cell Culture

The CCD-11Lu (CCL-202) cell line (normal human lung fibroblasts) was obtained from ATCC. Cells were cultured in DMEM (Dulbecco’s Modified Eagle Medium, high glucose) + GlutaMAX with the supplementation of penicillin (50 μg/mL), streptomycin (50 U/mL) and 10% FBS. Cells were grown at 37 °C and 5% CO_2_ and passaged twice before the experiment.

### 2.4. In Vitro Antimicrobial Activity Assay

The newly synthesized compounds **3a**–**3k** were investigated in vitro for antimicrobial activities. In these studies, the broth microdilution was used. The tests were performed in accordance with the guidelines of the European Committee on Antimicrobial Susceptibility Testing (EUCAST) [31] and Clinical and Laboratory Standards Institute (CLSI) [32]. All the used microbial cultures were first subcultured on nutrient agar (for bacteria) or Sabouraud agar (for fungi) (BioMaxima S.A., Lublin, Poland). The surface of Mueller-Hinton agar (BioMaxima S.A., Lublin, Poland) and RPMI (Roswell Park Memorial Institute) 1640 with MOPS (3-(*N*-Morpholino)propanesulfonic acid) (Sigma-Aldrich Chemicals, St. Louis, MO, USA) were inoculated with the suspensions of bacterial or fungal species, respectively. Microbial suspensions were prepared in sterile saline (0.85% NaCl) with an optical density of 0.5 McFarland standard scale. Samples containing examined compounds were dissolved in dimethyl sulfoxide (DMSO). The initial concentrations of these derivatives were 50 mg/mL in DMSO. In the next stage, all microbial suspensions were plated on solid media with 2 mg/mL of the tested compounds (diluted 25-fold in medium) and then incubated under specific conditions. The inhibition of bacterial and fungal growth was assessed by comparison with control cultures in media without any sample tested. Standard drugs—ciprofloxacin (antibacterial chemotherapeutic) and nystatin (antifungal antibiotic) (Sigma-Aldrich Chemicals, St. Louis, MO, USA)—were used as reference substances [31,32].

Furthermore, the MIC (minimal inhibitory concentration) of the new thiazoles was evaluated by the microdilution broth method in 96-well polystyrene plates. In this study, two-fold dilutions of the tested compounds in selective broth—Mueller-Hinton (BioMaxima S.A., Lublin, Poland) for bacteria and RPMI 1640 with MOPS (Sigma-Aldrich Chemicals, St. Louis, MO, USA) for fungi—were performed. The final concentrations of these compounds ranged from 1000 (diluted 50-fold in broth) to 0.0038 µg/mL. Bacterial and fungal suspensions were prepared in sterile NaCl with an optical density of 0.5 McFarland standard. Next, the microbial suspension was introduced into each well of the microplate containing broth and various concentrations of the tested substances. After 24 h incubation, the MIC value was assessed in the BioTek spectrophotometer (Biokom, Janki, Poland) as the minimal concentration of the samples that showed complete microbial growth inhibition. Appropriate DMSO, sterile and growth controls were performed. The media with and without tested substances/DMSO were used as controls [31,32,33,34].

Subsequently, MBC (minimal bactericidal concentration) or MFC (minimal fungicidal concentration) was determined by transferring the cultures from each MIC determination well to the appropriate solid medium. After incubation, the lowest compound concentrations with no visible growth observed were evaluated as a bactericidal or fungicidal concentration. All the experiments were repeated three times as independent assays, and representative data are presented [31,32,33,34].

### 2.5. Antibiofilm Activity Assay

*Candida albicans* ATCC 10231 biofilms were grown on the polystyrene flat-bottomed microtiter plates. The initial cell suspension was grown in RPMI 1640 medium until the cell density of 2 × 10^6^ cells/mL was attained and dispensed into the wells of two microtiter plates (100 µL per well). The effect of the substances **3b**, **3c** and **3e** on the biofilm-forming ability was verified in the presence of different concentrations (1/8 × MIC, 1/4 × MIC, and 1/2 × MIC). As a control, 1% DMSO was used. The plates were incubated for 48 h at 37 °C. Next, the medium was aspirated, and planktonic-phase cells were removed by washing three times with PBS (phosphate-buffered saline, pH 7.4). Biofilms were then dried at 60 °C for 2 h. Biofilm formation (based on its biomass) was determined using the crystal violet (CV) assay described by Feoktistova M. et al. [35], and its metabolic activity was determined with the use of a colorimetric XTT [2,3-bis(2-methoxy-4-nitro-5-sulfophenyl)- 2H-tetrazolium-5-carboxanilide sodium salt] reduction assay as previously reported [10,36]. The studies were performed using a spectrophotometer (the absorbance was measured at 570 nm and 490 nm in CV staining and XTT reduction assay, respectively).

For the mature biofilms, the cell suspension was grown in RPMI 1640 to the final cell density of 1 × 10^6^ cells/mL. A total of 100 µL of the cell suspension was dispensed into the wells and incubated at 37 °C for 2 days. Then, the nonadherent cells were gently washed with PBS. The wells were then filled with 100 µL of two-fold dilutions of tested compounds in RPMI 1640 corresponding to MIC, 2 × MIC and 8 × MIC. For the control, 1% DMSO was used. Furthermore, microtiter plates were incubated at 37 °C for 2 days. Subsequently, the appropriate incubation medium was aspirated from the wells and nonadherent cells were removed by washing the biofilms as described previously. The CV staining of dry biofilms and the colorimetric XTT reduction assay were performed according to method cited above. Each experiment was repeated three times as independent assays. All data are expressed as a mean ± SD (standard deviation) of three independent experiments. The differences between the biomass and the viability of biofilm cells were compared to the control (untreated cells) with a two-sided student’s *t*-test, using GraphPad Prism version 9.1.1. (San Diego, CA, USA). The *p* value < 0.05 was considered statistically significant.

### 2.6. Membrane Permeability Assay

The alteration of membrane permeability of *C. albicans* ATCC 10231 was evaluated using a crystal violet (CV) assay proposed by Lee HS et al. [37]. *C. albicans* cells at the exponential phase were harvested by centrifugation at 4500× *g* at 4 °C for 5 min. The cells were washed twice and resuspended in 0.85% NaCl. The tested compounds **3b**, **3c** and **3e** corresponding to the concentrations of MIC, 2 × MIC or 8 × MIC were added to the suspension and incubated at 37 °C, 200 rpm for 8 h. Solvent (DMSO) controls were included for each compound. Cells were harvested and washed in 0.85% NaCl, and the cell density was adjusted in each experimental group to equate to 1 × 10^8^ cells/mL. Next, the cells were resuspended in 0.85% NaCl supplemented with 10 μg/mL of CV, followed by incubation at 37 °C, 200 rpm for 10 min. The cells were then seeded by centrifugation at 12,000 × *g* at 4 °C for 15 min, and the remaining CV in the supernatant was measured at 590 nm. The OD values of the initial crystal violet solution used in the assay were regarded as 100%. The crystal violet uptake (%) was calculated as follows: uptake of crystal violet (%) = (A590 of the sample)/(A590 of crystal violet solution) × 100.

### 2.7. Sorbitol Assay

To investigate the effect of the newly synthesized compounds **3b**, **3c** and **3e** on the cell wall of *C. albicans* ATCC 10231, the sorbitol assay was used. The sorbitol (Sigma-Aldrich Chemicals, St. Louis, MO, USA) was added to the culture medium in a final concentration of 0.8 M. The MICs of the newly thiazole derivatives towards yeast, using Sabouraud Dextrose Broth (SDB) medium (BioMaxima S.A., Lublin, Poland) in the presence and absence of sorbitol (as control), were determined in different lines of the same microplate. The microdilution technique was performed in triplicate according to the guidelines of previous authors [7,38,39,40]. After filling each well of the microplates with 100 µL of SDB and 100 µL of SDB supplemented with sorbitol, serial dilutions of studied compounds and nystatin (as control) ranging from 0.03 to 1000 μg/mL were carried out. In the next stage, 10 µL of *C. albicans* suspension (10^6^ CFU/mL) was added to each well. The appropriate controls for yeast growth and sterility were also performed. MICs were read after 2 and 7 days incubation at 37 °C as the lowest concentrations of tested compounds capable of visually inhibiting the candidal growth.

### 2.8. Ergosterol Assay

To assess if the compounds **3b**, **3c** and **3e** bind to the fungal membrane sterols of *C. albicans* ATCC 10231, test with ergosterol was performed according to the procedure previously reported [7,38,39,40]. The stock solution of exogenous ergosterol (Sigma-Aldrich Chemicals, St. Louis, MO, USA) at a final concentration of 10 mg/mL in the liquid RPMI-1640 medium with DMSO (no more than 10% of final volume) with an addition of 1% Tween 80 (Pol-Aura, Różnowo, Poland) was prepared. The MICs of the newly formed derivatives towards yeast were determined by the broth microdilution method, with and without exogenous ergosterol added to the RPMI-1640 medium in different lines of the same microplate. The ergosterol was transferred to the wells in a final concentration of 400 μg/mL. After filling each well of the microplates with 100 µL of medium with and without ergosterol, serial dilutions of newly synthesized compounds and nystatin (as control) ranging from 0.03 to 1000 μg/mL were carried out. Then, 10 µL of yeast suspension (10^6^ CFU/mL) was added to each well. After 24-h incubation at 35 °C MIC was determined as the lowest concentration of tested compounds inhibiting the visible growth of *C. albicans*. Yeast growth and sterility were also controlled.

### 2.9. Cytotoxicity Assay

For cytotoxicity assay normal human lung fibroblasts were seeded in 96-well microplates at a density of 2.5 × 10^4^ cells/mL in 100 μL DMEM + GlutaMAX supplemented with 10% heat-inactivated FBS (Fetal Bovine Serum) in three sets for different periods of tested compound **3b**, **3c**, and **3e** exposure. After 24 h of cell attachment, plates were washed with 100 μL/well with phosphate buffered saline (PBS) and the cells were treated with five concentrations (0.25, 0.5, 1, 10 and 25 μg/mL final concentration) of each compound prepared in fresh FBS-free medium for 24, 48 and 72 h. Each concentration was tested in triplicate. All sets included wells containing 0.01% DMSO as a negative control. Cytotoxicity of compounds was assessed using the MTT-based colorimetric assay described below. The MTT is 3-(4,5-dimethylthiazol-2-yl)-2,5-diphenyl tetrazolium bromide), indicating the number of viable cells and the metabolic activity level in a given sample [41]. After 24, 48 and 72 h of compound exposure, the control and test media were removed, and the cells were rinsed with PBS and 100 μL of fresh medium (without FBS or antibiotics) was added to each well. A total of 10 μL of MTT (5 mg/mL) prepared in PBS was added to each well, followed by incubation for 3 h at 37 °C in a 5% CO_2_ humidified incubator. Next, the medium was discarded, the cells were rinsed with 100 μL of PBS and 100 μL of DMSO was added to each well for dye extraction. The plate was then shaken for 10 min, and the absorbance was measured at 570 nm. Viability was calculated as the ratio of the mean of OD obtained for each condition to the control condition. All viability data are expressed as means ± standard deviation (SD).

### 2.10. Hemolytic Activity Assay

The blood samples were collected in sterile tubes with citrate dextrose as an anticoagulant. Samples were centrifuged at 500× *g* for 10 min at 4 °C in order to separate erythrocytes. The erythrocytes were washed using PBS buffer (10 mM phosphate, pH 7.5; 150 mM NaCl) until a transparent supernatant was obtained. Next, the erythrocytes were resuspended in PBS buffer at a final concentration of 2%. The tested substances were prepared at MIC, 2 × MIC, 5 × MIC and 10 × MIC concentrations in a 50 μL of DMSO. Finally, the compounds were mixed with 450 μL of the 2% erythrocyte suspension, incubated for 1 h at 37 °C and centrifuged at 5000× *g* for 10 min. Absorbance was read at a wavelength of 415 nm [42,43].

### 2.11. Quantum Chemical Calculation

Quantum chemical investigation was performed for the three most active molecules, that is compounds **3b**, **3c** and **3e**, using the Gaussian 09 package [44]. Optimization of geometrical parameters of the investigated systems was carried out at the DFT approximation, employing the B3LYP exchange-correlation functional and the 6-311G**basis set. The resulting structures were confirmed to be real minima on potential energy surface through calculation of the corresponding vibrational frequencies at the same level of approximation. Next, the HOMO and LUMO orbitals as well as the MEP surfaces were obtained based on the B3LYP/6-311G** calculations employing the Gaussian 09 package [44] and visualized with GaussView 5 [45]. The most positive electronic potential is presented in the MEP surface plots in blue, and the most negative one in red.

## 3. Results

### 3.1. Chemical Synthesis

The desired (dicyclopropylmethylene)hydrazinyl)thiazoles (**3a**–**3k**) were readily prepared by a two-step reaction as shown in Scheme 1. The first step was the reaction of the appropriate dicyclopropyl ketone and thiosemicarbazide in anhydrous ethyl alcohol in the presence of acetic acid as catalyst. The 2-(dicyclopropylmethylidene) hydrazinecarbothioamide (**1**) was obtained in good yield (56%). Then, hydrazinecarbothioamide **1** was reacted with the appropriate bromoketones in ethyl alcohol and under reflux to produce target products **3a**–**3k** with good to excellent yields (32–99%). All compounds were purified by silica gel column chromatography and characterized by spectroscopic methods such as ^1^H, ^13^C NMR, and ESI-HRMS. In the ^1^H NMR spectrum of hydrazinecarbothioamide **1**, the three signals at 7.36, 8.00 and 10.00 ppm were assigned to the protons of the NH_2_ and NH groups, respectively. In the ^1^H NMR spectra of compounds **3a**–**3k**, a thiazole-5H proton signal at about (7.11–7.61) ppm and a broadened hydrazine NH signal at (10.96–11.17) ppm were observed. The conversion of the substrates to the target products was also confirmed by the ^13^C NMR spectra. The signals at (102.63–108.67) ppm correspond to the CH carbon atoms in the thiazole ring. Additionally, the signals from the C=N and C–NH groups can be observed at about 150 and 170 ppm. Moreover, their ESI-HRMS spectra showed peaks corresponding to their molecular [M + H]^+^ ions.

### 3.2. Antimicrobial Activity

In this study, no antimicrobial activity was defined as a MIC > 1000 µg/mL, mild activity as a MIC in the range 501–1000 µg/mL, moderate activity with MIC from 126 to 500 µg/mL, good activity as a MIC in the range 26–125 µg/mL, strong activity with MIC between 10 and 25 µg/mL and very strong activity as a MIC < 10 µg/mL. The MBC/MIC or MFC/MIC ratios were calculated in order to determine bactericidal/fungicidal (MBC/MIC ≤ 4, MFC/MIC ≤ 4) or bacteriostatic/fungistatic (MBC/MIC > 4, MFC/MIC > 4) effect of the tested compounds [33,34].

#### 3.2.1. Antibacterial Activity

On the basis of our data presented in Table 1 and Table 2, **3a**–**3k** compounds had some antibacterial activity especially against the reference strains of Gram-positive bacteria. *Bordetella bronchiseptica* ATCC 4617 from Gram-negative bacteria was also sensitive to these substances but the remaining Gram-negative bacteria were insensitive to all studied compounds. Among these substances, **3e, 3g** and **3k** had the widest spectrum of activity with MIC ranging from 31.25 µg/mL to 1000 µg/mL and MBC from 500 µg/mL to >1000 µg/mL with good (**3e** and **3k**), moderate (**3e**, **3g** and **3k**) or mild (**3e** and **3g**) effect. In turn, reference *B. bronchiseptica* strain was the most sensitive to all compounds (MIC = 31.25–125 µg/mL in the case of **3a**–**3e**; MIC = 250–500 µg/mL for **3f**, **3h**–**3k** and MIC = 1000 µg/mL in the case of **3g**).

The minimal concentrations of all compounds that killed these rods were ≥1000 µg/mL. The tested substances also exhibited a similar effect against *S. aureus* ATCC 25923 (except **3f**, which had no activity), *Staphylococcus epidermidis* ATCC 12228 (except **3c**), and *Micrococcus luteus* ATCC 10240 (except **3i**) (MIC = 62.5–1000 µg/mL and MBC ≥ 1000 µg/mL). The activity of compounds **3a**–**3k** towards other bacteria (*S. aureus* ATCC 6538 and sporulating bacilli—*Bacillus subtilis* ATCC 6633 and *Bacillus cereus* ATCC 10876) was lower—mild or moderate (MIC = 250—≥1000 µg/mL, MBC = 500—≥1000 µg/mL).

#### 3.2.2. Antifungal Activity

The studied compounds showed a differential effect towards the reference fungal strains belonging to *Candida* spp. (Table 3 and Table 4). Among them, **3a**–**3f** and **3k** substances showed very strong activity with MIC = 0.24–7.81 µg/mL and MFC = 0.48–62.5 µg/mL against *Candida* spp. strains, except *C. glabrata* ATCC 90030. The **3g** and **3h** compounds also exhibited a high activity with a very strong or strong effect towards these microorganisms (MIC = 7.81–15.62 µg/mL and MFC = 15.62–62.5 µg/mL). *C. glabrata* ATCC 90030 was less sensitive to the above substances. The minimal concentrations of **3a–3e** and **3h** compounds which inhibited growth of these strains were 62.5 µg/mL. The highest effect towards *C. glabrata* was shown by **3f** with MIC = 31.25 µg/mL. In turn, the MICs of **3g** and **3k** substances were 125 µg/mL. These compounds **3a**–**3h** and **3k** showed good activity against *C. glabrata* and their MFC ranged from 125 to >1000 µg/mL. The lowest activity against all reference *Candida* spp. Exhibited two studied compounds: **3i** and **3j** with MIC = 125–500 µg/mL and MFC ≥ 1000 µg/mL.

In general, most of the newly synthesized compounds **3a**–**3k** indicated a fungicidal effect (MFC/MIC = 2–4), while the substances **3f**, **3i**, **3j** and **3k** had a fungicidal or fungistatic effect (MFC/MIC > 4).

#### 3.2.3. Effect on *Candida albicans* Biofilm

Due to the high antifungal activity of the most of novel compounds, the effect of compounds **3b**, **3c** and **3e** on the biofilm of the reference strain *C. albicans* ATCC 10231 was also assayed. The antibiofilm activity was studied by measuring the biofilm biomass and its metabolic activity after treatment with the abovementioned compounds. The effect on the biofilm-forming ability (and inhibition of the adhesion phase) was tested in the presence of three different concentrations (1/8 × MIC, 1/4 × MIC, and 1/2 × MIC). The biomasses of biofilms were determined by a crystal violet (CV) assay, which is one of the most popular methods to detect maintained adherence of cells. It is based on the staining of attached cells with CV dye, which binds to their proteins and DNA [35,46]. Cells that undergo cell death, under the influence of antifungals, lose their adherence and are subsequently lost from the population of cells, reducing the amount of CV staining in a culture [35]. The results showed that the studied compounds in the used concentrations (≤1/2 × MIC) have no inhibitory effect on the adhesion phase and biofilm formation by the reference *C. albicans* ATCC 10231 strain. The obtained OD values for each condition were comparable to the control (OD ≈ 1.05).

Moreover, colorimetric tetrazolium salt (XTT) reduction assay was undertaken to examine the effect of newly synthesized compounds on viability of *C. albicans* cells within the forming biofilm and the metabolic activity of this structure. It relies on the reduction of yellow tetrazolium salt XTT by dehydrogenase enzymes of metabolically active cells yielding an orange-coloured, water-soluble formazan [47]. Live cells reduce the tetrazole ring and formazan is formed, which can be assessed visually and quantified spectrophotometrically [48,49]. The obtained results also confirmed the absence of an effect of these compounds on the metabolic activity of the forming biofilm of *C. albicans* ATCC 10231 at the same concentrations that were used in the CV assay. In this study obtained OD values for three concentrations of these compounds were also comparable with the control (OD ≈ 1.75).

The antibiofilm effect of these compounds was also tested towards the mature biofilm of the reference *C. albicans* strain. Biofilm biomass and the cellular metabolic activity were studied using the same methods as previously employed, by CV staining and XTT reduction assay, respectively. It was shown that the newly synthesized derivatives have no eradication effect against mature biofilms at the concentration values corresponding to an MIC 2 times and 8 times higher than the MIC for planktonic cells (OD ≈ 0.55 were comparable with the control), consistent with the prior observations that mature biofilms are more resistant to antifungal drugs [11].

In the case of metabolic activity assay (Table 5), some very slight changes were identified on mature biofilm for compound **3b**. A slight decrease in this activity (OD = 1.76 ± 0.19) was observed at the higher concentration (eightfold higher than MIC for planktonic cells). In the case of remaining compounds, OD obtained for selected MIC value (MIC, 2 × MIC and 8 × MIC) was comparable to the control.

#### 3.2.4. Membrane Permeability Assay

Alteration of membrane permeability of *C. albicans* ATCC 10231 was also investigated using a CV assay (Table 6). Crystal violet, which generally poorly penetrated the membrane, could enter the damaged cell membrane. Cells treated with compounds were stained, indicating changes in membrane permeability and cell death. In contrast, control cells, not treated with tested compounds, did not stain with CV, indicating live cells with intact cell membranes [50]. In the case of compounds **3b**, **3c**, and **3e**, an enhancement in the uptake of CV was observed. These compounds in higher concentrations increase the permeability of the membrane/cell wall of *C. albicans* strains and take up CV from the medium. The highest CV uptake (28%) was shown for compound **3c** at a concentration eight times higher than MIC for planktonic cells. In the case of compounds **3b** and **3e**, CV uptake was 11.4% and 1.8%, respectively.

#### 3.2.5. Mode of Action with Sorbitol and Ergosterol Assay

The mode of action of the newly synthesized compounds **3b**, **3c**, and **3e**, presented in Table 7, was tested in order to define whether their anticandidal effect involved a direct interaction with the cell wall structure of *C. albicans* (via testing with sorbitol) and/or with the ion permeability of the membrane of this organism (via the test with ergosterol) [7,38,39,40,51]. Sorbitol has an osmoprotectant function and was used to stabilize fungi protoplasts. Sorbitol protects yeast cells, which can grow in the presence of compounds that are inhibitors of their cell wall. As a result, the MIC value increases in the medium supplemented with sorbitol. In turn, in the sorbitol-free medium, Candida growth is inhibited and the MIC decreases [7,38,39,40,51].

The obtained results showed that there are some differences between the MIC values of the tested compounds against the reference *C. albicans* strains in either the absence or presence of sorbitol. MIC values of compounds **3b** and **3e** for *C. albicans* ATCC 10231 strain increased 8-fold and even 32-fold for compound **3c** in the presence of sorbitol. Therefore, we can suppose that the mode of action of the newly synthesized compounds may be related to the influence on the cell wall structure of *C. albicans*. However, this requires further research.

The potential influence of these compounds on the fungal cell membrane was also evaluated. Therefore, the ability to form complexes with the main fungal sterol—ergosterol—was investigated. The activity of the newly formed thiazole derivatives may be also caused by binding to fungal sterol, because their MIC values in the medium with exogenous ergosterol were 32 times or 64 times higher compared with the ergosterol-free experiment. The exogenous ergosterol prevents the binding to ergosterol in the fungal membranes. Consequently, the MIC increase for these compounds occurs because only increased product concentrations in the medium might assure interaction with ergosterol in the fungal membranes [7,38,39,40].

As can be seen, the MIC values of nystatin (used as control) in the media with and without sorbitol, were identical, suggesting that this antibiotic does not act by inhibiting fungal cell wall synthesis. In turn, 16-fold increase in the MIC values after the addition of ergosterol was observed. This indicates that the mechanism of action of nystatin involves complexation with ergosterol, which is well known and consistent with the data in the literature.

### 3.3. Cytotoxic Activity

In the next stage of our research, the toxicity of the newly synthesized compounds was assayed. This is a very important study that determines whether the tested compounds will be suitable for further clinical investigation [28]. To demonstrate that these compounds are safe for host cells, we decided to investigate their cytotoxic effect on normal human lung fibroblasts using an MTT assay. In this experiment CCD-11Lu cell lines were exposed to five different concentrations of tested compounds **3b**, **3c** and **3e** for 24 h, 48 h and 72 h. The obtained results indicate that these compounds do not show significant cytotoxic properties towards human lung fibroblasts. The cell viability percentage was shown in Table 8. It was observed that the viability of normal cells after 24, 48 and 72 h exposure to the tested compounds at concentrations of 0.25, 0.5, 1, 10 and 25 µg/mL varied slightly with time and dose and was similar. However, the highest mean cell viability percentage was with the lowest substance concentration (about 100% viability at 0.25 µg/mL). In turn, in the case of compound **3b** at a concentration of 25 µg/mL, cell viability was 86%. These results confirm that the antifungal effect of the new compounds against the reference *C. albicans* strains was observed at their noncytotoxic concentrations.

### 3.4. Hemolytic Activity

In the present study, the erythrocyte model was used to assess the hemolytic activity of compounds **3b**, **3c** and **3e**. Hemolysis was due to red blood cell destruction, which resulted from the lysis of their membrane lipid bilayer [42,43]. To estimate the relative hemolytic potential of the tested compounds, the appropriate controls: 100% erythrocyte lysis using 1% Triton X-100 (positive control) and no lysis in DMSO (negative control) were used. As presented in Table 9, the concentrations with a value of MIC, 2-fold, 5-fold and 10-fold higher than MIC of the studied compounds not exerting any hemolytic effects. The percentage of lysed red blood cells ranged from 0.01 to 1.11. In these concentrations, the compounds are not toxic to erythrocytes. The present results showed, like the cytotoxicity studies, that the new compounds indicated anticandidal effect at their non-cytotoxic concentrations.

### 3.5. Quantum Chemical Calculation

The results of quantum chemistry calculations are presented in Table 10, the shape of HOMO and LUMO orbitals in Figure 1, and the MEP surfaces in Figure 2. The HOMO orbitals have a very similar shape for all three compounds, in each case covering almost the whole molecule. An analogous situation is observed for the LUMO orbitals. The values of the HOMO–LUMO energy gap are in the order of 4.2–4.3 eV for the three investigated compounds. The energy of the HOMO and the LUMO is 0.2 and 0.3 eV, respectively, higher for the methyl derivative than for the remaining two derivatives. Consequently, the **3e** molecule has the lowest values of ionization potential IP, electron affinity EA, and electronegativity χ. Thus, among the investigated systems, electron removal is the easiest for compound **3e**. Contrary, compounds **3c** and **3b**—having larger values of EA and larger electronegativity—are the two best electron acceptors. They are also the compounds with the largest values of softness (the lowest values of hardness η), and thus the most active among the investigated systems.

In Figure 2, the electron-rich regions of the MEP surfaces are denoted in red, and those electron deficient in blue. Their analysis shows that the electron-rich regions are located around nitrogen atoms in all three systems. An additional electron-rich region is located around the chloro substituent in compound **3b** and around the bromo substituent in compound **3c**. As the methyl substituent is an electron-donating group, it increases electron density in the phenyl ring, which is reflected by the MEP of **3e** molecule. The electron-deficient regions are located around cyclopropyl rings.

## 4. Discussion

It is worth noting that the treatment of candidiasis, an especially invasive disease, is often ineffective, since the list of antifungals is very limited, while many of them have been extensively used, leading to the development of antifungal resistance [52]. Therefore, there is a strong need to search for new antifungals. Prompted by the abovementioned fact, in this research, the novel dicyclopropyl-thiazole compounds were designed, synthesized and characterized by physicochemical parameters together with their investigation for in vitro antimicrobial effect.

The obtained results showed that most of the studied novel compounds possessed very a strong antifungal effect (MIC = 0.24–7.81 µg/mL). The *Candida* spp. strains were especially sensitive to compounds **3a**–**3f** and **3k** (with the exception of *C. glabrata* ATCC 90030). Moreover, almost all substances showed a fungicidal effect (MFC/MIC = 2–4), which is very beneficial from a therapeutic point of view. Due to the high antifungal activity of these compounds, some of them, namely **3b**, **3c**, and **3e** were used for further detailed research. As the predominant cause of all types of candidiases is *Candida albicans,* for these studies reference *C. albicans* ATCC 10231 strain was used. The antibiofilm activity and mode of action together with the effect on membrane permeability in *C. albicans* were investigated.

It is known that fungal cell adhesion and biofilm formation are the key points for the pathogenesis and antifungal treatment [49]. The ability to evaluate the compounds’ activity during the growth of biofilm and against mature biofilm is interesting for the investigation of resistance mechanisms and novel antifungal therapies [10,47]. The obtained results showed that newly synthesized dicyclopropyl-thiazole derivatives, although they had a strong antifungal effect, have no inhibitory effect (at 1/2 × MIC) or eradication effect (at 8 × MIC) towards the formation of biofilm and the mature structure, respectively. Moreover, only for compound **3b**, a slight decrease in the metabolic activity of mature biofilm (at an eightfold higher MIC) was observed. It is noteworthy that cells grown in the biofilm community are much more resistant to noxious agents than the nonbiofilm, planktonic cells. From a medical point of view, the particular feature of biofilm-forming yeasts is their insensitivity to antimycotics. The mature biofilm presents a therapeutic challenge in the management of device-associated *Candida* infections. Therefore, the development of new antifungal agents with new drug targets for the treatment of biofilm-associated infections is very important [10,39].

Studying the potential mechanisms of the action of the compounds **3b**, **3c** and **3e**, it was of interest to check whether their antifungal activity involved a direct interaction with the cell wall structure of *C. albicans* and/or with the ion permeability of the membrane of this organism [38]. The sorbitol assay is based on this compound acting as an osmoprotector which stabilizes fungi protoplasts, and in consequence, the antifungal effects of cell wall inhibitors are reversed in media containing sorbitol. This effect is detected by the increase in the MIC value observed in medium with sorbitol as compared to the MIC value in medium without sorbitol. The obtained results showed that there were some differences between MIC of the tested compounds against reference *C. albicans* strains in the absence or presence of sorbitol. MIC values of compounds **3b** and **3e** for reference *C. albicans* strain increased 8 times and even 32 times for compound **3c** in the presence of sorbitol. These data suggest that the mode of action of the novel thiazole derivatives may be related to action within the fungal cell wall structure.

Moreover, given the possible interference of these compounds with fungal cell membranes, their ability to form complexes with ergosterol was investigated. Ergosterol as a major sterol component present in the plasma membrane of fungi and plays the same role in fungal membranes that cholesterol plays in mammalian cells [7,38,39,40]. The activity of the novel thiazole derivatives may also be caused by the binding to fungal sterol, because their MIC values in the medium with exogenous ergosterol were 32-fold or 64-fold higher compared with the ergosterol-free medium. The exogenous ergosterol prevents the binding to ergosterol in the fungal membranes. Consequently, an MIC increase for these compounds occurs because only increased product concentration in the medium might assure interaction with ergosterol in the fungal membranes [7,38,39,40]. In the case of nystatin used as control agents with a known mechanism of action involving complexation with ergosterol, a 16-fold increase in the MIC values in the presence of ergosterol was shown.

In our research, the alteration of membrane permeability of yeasts after treatment with tested compounds was also determined. Membrane permeability was investigated by the uptake of CV that only penetrated compromised cell membranes, resulting in cell staining. In turn, live cells with intact cell membranes, not treated with tested compounds, did not stain with CV. The highest CV uptake (28%) was shown for compound **3c** at a concentration eight times higher than MIC.

The cytotoxicity and safety assessment of potential medicinal components are very important parameters that also need to be evaluated. Therefore, the next stage of this study was to determine in vitro cytotoxicity effect of the selected compounds, namely **3b**, **3c** and **3e** on normal human lung fibroblasts using MTT test. Cell culture techniques are useful for the evaluation of the toxicity of different compounds or materials. The results obtained from these in vitro assays might be indicative of the effects observed in vivo [53]. Among in vitro assays, cytotoxicity tests are simple, reproducible, inexpensive to perform, and suitable for evaluation of basic biologic properties [41]. These studies also provide a significant amount of information can be conducted under controlled conditions and may elucidate the mechanisms of cellular toxicity [41,54,55]. Several methods are available for cytotoxicity testing and among them, the MTT-based colorimetric assay is a standard method. This test indicates the number of viable cells and the level of metabolic activity in a sample [41]. The concentrations of the tested compounds and exposure time are very important parameters in cytotoxicity assay. According to Basak et al. [41], the absorption value obtained with the control cells was adjusted to 100% viability. Cytotoxicity was rated based on cell viability relative to the control group: non-cytotoxic > 90%, slightly cytotoxic 60–90%, moderately cytotoxic 30–59% and severely cytotoxic < 30% cell viability. In turn, according to data presented by Kazak et al. [54], cell viability lower than 70% indicates that the material has cytotoxic potential. In present study, no cytotoxicity (cell viability > 90%) was found for all concentrations of the tested compounds even after exposure for 72 h. The highest viability percentage for normal human lung fibroblast cells (100%) was found in the lowest concentrations of compounds (0.25 µg/mL). In turn, at a concentration of 25 µg/mL of compound **3b** after 24 h exposure, 86.45% cell viability was observed. On the basis of these results, no cytotoxic activity of this compound can be confirmed.

In the present studies, red blood cells (RBCs) were also used to assess the influence of the compounds **3b**, **3c** and **3e** on their cell membranes. The lysis of the membrane lipid bilayer of RBCs is related to the concentration and potency of the studied agents. Hemolysis refers to the disruption of RBCs, and the assay detects the leaking of intracellular contents including hemoglobin. The extent of hemolysis was estimated from the amount of the extracellular hemoglobin [7,56]. Many studies have reported that in vitro haemolysis assays have a good correlation with in vivo toxicity. We observed no hemolytic effect of these compounds at concentrations up to 10 times higher than MIC. The erythrocyte lysis assay also confirmed that the newly synthesized derivatives showed an antifungal effect at their non-cytotoxic concentrations towards the reference *C. albicans* strains. The similar antifungal activity of **3b**, **3c** and **3e** compounds was confirmed by their similar Δ*E* values.

## 5. Conclusions

The obtained results showed that most of the newly synthesized dicyclopropyl-thiazole derivatives indicated a strong antifungal effect at non-cytotoxic concentrations against the reference *Candida* spp. strains. These compounds increased the permeability of the membrane/cell wall of yeasts, and their mode of activity may be related to action within the fungal cell wall structure and/or within the cell membrane. It was observed that their antifungal action is not related to effects on the biofilm structure or its metabolic activity. These compounds represent a very promising group of antifungals for further preclinical studies.

## Data Availability

Data sharing not applicable.

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
