# Peer review of "Synthesis and Physicochemical Characterization of Novel Dicyclopropyl-Thiazole Compounds as Nontoxic and Promising Antifungals"

_materials, 2021, doi:10.3390/ma14133500_

Round 1

Reviewer 1 Report

This is an interesting and relatively complete study. However, there are some points to be improved:

-imine compounds are usually characterized to have a relatively low stability, and therefore it would be relevant to evaluate the stability of these compounds under different pH

-unfortunately MIC values are relatively higher when compared with positive control

-it would be interesting to evaluate the effects of these compounds on 14alpha-desmetilase enzyme, as previously it was described the CYP inhibition effect of thiazole systems (e.g. in ritonavir)

-the authors performed repetitions in several assays. Therefore, some statistical studies can be performed. In addition, for example in antibiofilm studies, the results (e.g. Table 6) seemed to have not considered the repetitions. Moreover, in cytotoxicity studies in fibroblasts the authors presents cell viability % plus variability – this is SD? SEM? Other? Therefore, statistics must clearly be better considered and described and applied in dada analysis in the whole manuscript

Author Response

Dear Reviewer 1

Thank you very much indeed for your kind comments and the time spent on the revision of our manuscript. Here below we present a point-by-point list of the responses to the obtained comments, in red.

This is an interesting and relatively complete study. However, there are some points to be improved:

Imine compounds are usually characterized to have a relatively low stability, and therefore it would be relevant to evaluate the stability of these compounds under different pH

Response: There is a number of articles available in the literature showing that hydrazones are stable under physiological conditions and unstable in strongly acidic and alkaline environments. Due to the planned use of these compounds as drugs, in our research we are interested in their behaviour under the conditions close to physiological. Under such conditions, we did not notice a spontaneous hydrolysis of the products. A time-consuming evaluation of the stability of these compounds under different conditions is unnecessary and not in line with the presented research.

Unfortunately MIC values are relatively higher when compared with positive control

Response: Compounds 3b, 3c and 3e have been shown to have comparable activity to nystatin. In addition, these compounds, compared to nystatin, are very small molecules, which are also easy to synthesize and do not show toxicity.

It would be interesting to evaluate the effects of these compounds on 14alpha-desmetilase enzyme, as previously it was described the CYP inhibition effect of thiazole systems (e.g. in ritonavir)

Response: We agree with the Reviewer that studies of the inhibitory properties of thiazoles towards 14-alpha-demethylase would be very interesting. However, such research requires the development of an appropriate methodology for testing such derivatives, which would be very time-consuming and the obtained results are uncertain. We will look at this idea, which may become the subject of our research in the future.

The authors performed repetitions in several assays. Therefore, some statistical studies can be performed. In addition, for example in antibiofilm studies, the results (e.g. Table 6) seemed to have not considered the repetitions. Moreover, in cytotoxicity studies in fibroblasts the authors presents cell viability % plus variability – this is SD? SEM? Other? Therefore, statistics must clearly be better considered and described and applied in dada analysis in the whole manuscript

Response: We would like to thank the Reviewer for his critical comments. All the mentioned shortcomings regarding the statistical analysis of biological studies were included in the manuscript.

On behalf of the Authors

Anna Biernasiuk

Reviewer 2 Report

This study evaluates the antimicrobial effects of some novel antimicrobials the authors synthesized in-house. While the study has extensive amount of chemical synthesis and antimicrobial testing, there are certain aspect of antimicrobial testing that need to be addressed:

  1. Please mention the total DMSO concentration in the assays as DMSO itself can have antimicrobial effect.
  2. Why was only sorbitol was chosen to address cell wall health? Other stressors need to be tested to totally assess the effect on the cell wall (for example..Eukaryot Cell. 2013 Feb; 12(2): 254–264.). Another way to do this would include testing synergy between these compounds and cell wall damaging antifungals like caspofungin. Or the authors can pre-expose the cells to their compounds and assess the sensitivity to cell wall damaging antifungals. Also, the assay conducted as is does not imply that the novel antifungals bind the cell wall, as concluded in the study (line 594).
  3. Please provide data corresponding to lines 529-552 and 560-564. Data must be shown to conclude the lack of effect.
  4. Mechanism of “antifungal” “Nystatin” is known (lines 607-611).

Author Response

Dear Reviewer 2

Thank you very much indeed for your kind comments and the time spent on the revision of our manuscript. Here below we present a point-by-point list of the responses to the obtained comments, in red.

This study evaluates the antimicrobial effects of some novel antimicrobials the authors synthesized in-house. While the study has extensive amount of chemical synthesis and antimicrobial testing, there are certain aspect of antimicrobial testing that need to be addressed:

Please mention the total DMSO concentration in the assays as DMSO itself can have antimicrobial effect.

Response: Samples containing the newly synthesized compounds were first dissolved in dimethyl sulfoxide (DMSO). The initial concentrations of these derivatives were 50 mg/mL in DMSO. In turn, their final concentrations in the media, in 96-well polystyrene plates, ranged from 1000 to 0.0038 µg/mL. Thus, the new compound (diluted 50-fold in broth) was introduced into the first well of the titration plate, and then it was two-fold serially diluted. The DMSO concentration was too low to inhibit microbial growth. The medium with serially diluted DMSO and without tested substance was also used as control.

Why was only sorbitol was chosen to address cell wall health? Other stressors need to be tested to totally assess the effect on the cell wall (for example..Eukaryot Cell. 2013 Feb; 12(2): 254–264.). Another way to do this would include testing synergy between these compounds and cell wall damaging antifungals like caspofungin. Or the authors can pre-expose the cells to their compounds and assess the sensitivity to cell wall damaging antifungals. Also, the assay conducted as is does not imply that the novel antifungals bind the cell wall, as concluded in the study (line 594).

Response: We agree with the Reviewer that other stressors (for example: prolonged high-temperature stress) need to be also tested to totally assess the effect on the cell wall. The topic is very interesting and we will carry out such research in the future, after developing an appropriate methodology. In our evaluation (mode of antifungal action), an osmotic protector – 0.8 M sorbitol was used. According to the literature data [References: 39-42], it is recommended to assess the effect on the cell wall. It was used during antimicrobial assays in order to check differences on MIC values of tested compounds (in the presence or absence of sorbitol). If the MIC of tested substance does not change – the mechanism of action is not involved with osmotic pressure and cell wall degradation. However, if the MIC value increases in the presence of sorbitol – tested substance interferes with osmotic pressure and causes degradation of the cell wall.

The aim of our further research is also the assessment of interactions between these derivatives and the selected antimycotics (as suggested by the Reviewier) using the fractional inhibitory concentration index (FICi). The our preliminary studies showed additive effect in the case of combination of tested compounds 3b, 3c, and 3e with eugenol (a natural component of essential oils, which disturb  the cell  wall  integrity). Moreover, their combinations with nystatin (a natural polyene antibiotic that interacts with membranes containing ergosterol) and chlorhexidine (a synthetic antiseptic, which impairs the integrity of the cell wall) were found to be noninterfering and these interactions were considered as indifferent. These studies are preliminary, very time-consuming and will be continued and extended to include new antifungals (e.g. caspofungin, fluconazole, posaconazole, selected antiseptics and natural components).

Our studies are preliminary, we can only assume that mode of action of these compounds may be related with the influence on the structure of the fungal cell wall and/or the cell membrane.

Please provide data corresponding to lines 529-552 and 560-564. Data must be shown to conclude the lack of effect.

Response: In these studies, no effect of the newly synthesized compounds (in three different concentrations: 1/8 x MIC, 1/4 x MIC, and 1/2 MIC) on the biofilm-forming ability, viability of C. albicans cells within the forming biofilm and the metabolic activity of this structure, was shown. Moreover, these new derivatives have no eradication effect against mature biofilm of reference C. albicans strain (at the concentration values: MIC, 2 x MIC, and 8 x MIC). In all cases, the obtained OD values for these compounds were comparable with the control. We did not include this data in tables/figures, but only in the form of text.

Mechanism of “antifungal” “Nystatin” is known (lines 607-611).

Response: We agree with the Reviewer that mechanism of the antifungal activity of nystatin is well-known. Therefore, we used it as a positive control in these studies (mode of antifungal action). Our results indicated an increase in the MIC value of nystatin in a medium supplemented with ergosterol, which confirms that its mode of action is related to the interaction with ergosterol-containing yeast membranes. These results are consistent with literature data.

On behalf of the Authors

Anna Biernasiuk

Reviewer 3 Report

The manuscript materials-1226403 "Synthesis and physico-chemical characterization of novel dicyclopropyl-thiazole compounds as non-toxic and promising antifungals" by Biernasiuk et.al. describes the synthesis of new series of dicyclopropyl-thiazole compounds and the study of their antibiofilm and antimicrobial activity. The obtained compounds have weak antibacterial properties, but they show high antifungal activity.

The search for new antimicrobial drugs is an important problem due to the resistance of microorganisms to existing drugs. In this manuscript, the design of antifungal agents based on thiazole derivatives was implemented. The manuscript is well written and logically structured. The paper will definitely be of interest to the readers of Materials. The level of English is sufficient to understand the text of the manuscript. The disadvantage of the manuscript is the lack of additional materials with the spectra of new compounds.

Questions and comments:

1) Line 73 "… as well as anti-SARS-CoV-2 [24]..." The results of the 2013 study [24] refer to SARS-CoV, not SARS-CoV-2.

2) Part 3.2. It is not clear why the calculations were made for only three compounds. And how did the authors choose them? The data on the best antifungal activity were written much later in the manuscript.

3) Since almost all compounds were obtained for the first time, new compounds should be characterized by few physical methods (1H, 13C NMR, IR spectroscopy and mass spectrometry). Images of all spectra should be added in supplementary materials.

Author Response

Dear Reviewer 3

Thank you very much indeed for your kind comments and the time spent on the revision of our manuscript. Here below we present a point-by-point list of the responses to the obtained comments, in red.

The manuscript materials-1226403 "Synthesis and physico-chemical characterization of novel dicyclopropyl-thiazole compounds as non-toxic and promising antifungals" by Biernasiuk et.al. describes the synthesis of new series of dicyclopropyl-thiazole compounds and the study of their antibiofilm and antimicrobial activity. The obtained compounds have weak antibacterial properties, but they show high antifungal activity.

The search for new antimicrobial drugs is an important problem due to the resistance of microorganisms to existing drugs. In this manuscript, the design of antifungal agents based on thiazole derivatives was implemented. The manuscript is well written and logically structured. The paper will definitely be of interest to the readers of Materials. The level of English is sufficient to understand the text of the manuscript. The disadvantage of the manuscript is the lack of additional materials with the spectra of new compounds.

Questions and comments:

Line 73 "… as well as anti-SARS-CoV-2 [24]..." The results of the 2013 study [24] refer to SARS-CoV, not SARS-CoV-2.

Response: Our mistake has been corrected.

Part 3.2. It is not clear why the calculations were made for only three compounds. And how did the authors choose them? The data on the best antifungal activity were written much later in the manuscript.

Response: The calculations were carried out for the three most active compounds, as described in the first sentence of section 2.9. To avoid any confusion, section 3.2 was now shifted to the end of the Results, and all tables numbering was properly updated.

Since almost all compounds were obtained for the first time, new compounds should be characterized by few physical methods (1H, 13C NMR, IR spectroscopy and mass spectrometry). Images of all spectra should be added in supplementary materials.

Response: Copies of the 1H, 13C NMR and HRMS spectra for all products are given in the Supplementary materials.

On behalf of the Authors

Anna Biernasiuk

Round 2

Reviewer 3 Report

I would like to thank the authors for improving the manuscript.